# How Active Inference Could Help Revolutionise Robotics

**DOI:** 10.3390/e24030361

**Published:** 2022-03-02

**Authors:** Lancelot Da Costa, Pablo Lanillos, Noor Sajid, Karl Friston, Shujhat Khan

**Affiliations:** 1Department of Mathematics, Imperial College London, London SW7 2AZ, UK; 2Wellcome Centre for Human Neuroimaging, University College London, London WC1N 3AR, UK; noor.sajid.18@ucl.ac.uk (N.S.); k.friston@ucl.ac.uk (K.F.); 3Department of Artificial Intelligence, Donders Institute for Brain, Cognition and Behavior, Radboud University, 6525 XZ Nijmegen, The Netherlands; p.lanillos@donders.ru.nl; 4Milton Keynes Hospital, Oxford Deanery, Milton Keynes MK6 5LD, UK; shujhat.khan15@imperial.ac.uk

**Keywords:** free energy, model-based control, adaptive robots, generative model, Bayesian inference, filtering, neurotechnology

## Abstract

Recent advances in neuroscience have characterised brain function using mathematical formalisms and first principles that may be usefully applied elsewhere. In this paper, we explain how active inference—a well-known description of sentient behaviour from neuroscience—can be exploited in robotics. In short, active inference leverages the processes thought to underwrite human behaviour to build effective autonomous systems. These systems show state-of-the-art performance in several robotics settings; we highlight these and explain how this framework may be used to advance robotics.

## 1. Active Inference

Active inference (AIF) is a unifying framework for describing and designing adaptive systems [1,2,3,4]. AIF emerged in the late 2000s as a unified theory of brain function [5,6] derived from statistical physics [2,7] and has since been used to simulate a wide range of behaviours in neuroscience [1,8], machine learning [9,10,11,12,13] and robotics [14]. AIF is an interesting framework for robotics because it unifies state-estimation, control and world model learning as inference processes that are solved by optimising a single objective functional: a free energy (also known as negative evidence lower bound), as used in variational Bayesian inference [15]. Furthermore, it endows robots with adaptive capabilities central to real world applications [14] (e.g., adaptation to internal and external parameter changes [16]). Additionally, its strong neuroscience foundation reduces the gap between engineering and the life sciences, thereby finessing human-centred robotic applications.

Although AIF has yet to be scaled—to tackle high dimensional problems—to the same extent as established approaches, such as deep reinforcement learning [17,18], numerical analyses generally show that active inference performs at least as well in simple environments [9,19,20,21,22,23], and better in environments featuring volatility, ambiguity and context sensitivity [21,22]. In this paper, we consider how AIF’s features could help address key technical challenges in robotics and discuss practical robotic applications. Our exposition provides a broad perspective that suppresses mathematical details, which can be found in the references herein [1,2,3,4,14,24,25].

In AIF, a generative model encodes an agent’s predictions (i.e., posterior beliefs), and preferred state and observation trajectories (i.e., prior beliefs) [2]. Behaviour realises the agent’s preferences by matching posterior with prior beliefs. Specifically, state-estimation, control and learning are unified by minimising a free energy functional scoring the discrepancy between current beliefs and prior preferences under the state-space model. For continuous states, AIF filters incoming observations through variational inference in generalised coordinates of motion [26]. This enables flexible and scalable inference algorithms and extends Kalman filters by accommodating non-linear, non-Markovian time-series [26,27,28]. AIF generalises discrete and continuous optimal control [29], and planning to partially observed environments, similarly to model predictive control or control as inference [30,31]. However, a crucial difference is that the (expected) free energy optimised during planning combines exploitative and explorative behaviour [32] in a Bayes optimal fashion [2,7]. The agent’s model—i.e., representations and goals—can then be learnt through few-shot learning [21], structure learning, imitation learning, and evolutionary approaches [1,33,34,35].

## 2. Solutions to Technical Challenges in Robotics

Current AIF models can help address challenges that require online adaptation, robustness and explainability, and may bring new perspectives to the state-of-the-art in estimation, control and planning—see Figure 1.

**Accurate and robust state tracking**. Filtering schemes developed for neuroimaging time-series [26] enable accurate state-tracking in highly complex and volatile environments [27,36]. This allows for continuous refinement of past, present, future state-estimation and the estimated precision of sensors as new information arrives [37] (c.f., Bayes optimal estimators of Kalman gain [38]). Moreover, AIF fuses multiple sensory streams, by weighing incoming sensory information by their estimated precision [36,39]. This enables accurate and robust inferences.**Adaptive model-based and shared control.** Describing the agent’s behaviour with a generative model—prescribing attracting states and trajectories—ensures robustness and adaptivity in the presence of noise, external fluctuations, and parameter changes. AIF humanoid robots [36] and industrial manipulators [40] show improved behaviour in the presence of internal and external parameter changes [16] and shared compliance control [41]. The robot’s autonomy—in shared control—can also be dynamically tuned. In particular, the operator may be given high-level control and the robot low-level control.**Learning and grounding.** AIF agents learn from sparse and noisy observations by actively sampling informative data points, enabling few-shot learning. Learning latent structure by optimising model evidence, subject to prior preferences in the generative model, leads to organising knowledge in hierarchical, sparsely interconnected modular (i.e., factorised) representations with temporal depth, usually represented with a graphical model [2]. This offers a promising pathway for biologically plausible neurosymbolic technologies [42,43].**Operational specification, safety and explainability**. AIF behaviour is explainable as a mixture of information and goal-seeking policies that are explicitly encoded (and evaluated in terms of expected free energy) in the generative model as priors—which can be specified by the user. Planning, which proceeds by generating counterfactual actions and assessing their consequences [1], can be monitored online and control can be returned to the user if necessary (i.e., policy switching). Moreover, the generative model can be specified as a directed graph (i.e., a Bayesian network), which entails the causal relationships between agent’s representations [44,45]. This affords an explicit and transparent explanation of sentient behaviour.

## 3. Practical Perspectives

Based on these properties, we envisage important applications of AIF in robotics.

**Context adaptive robots**. AIF agents build generative (world) models by continuously optimising free energy with regard to incoming data. This optimisation process maximises model accuracy while minimising complexity, which enables generalisation and context-adaptivity [36]. Contrariwise, robots that solely optimise accuracy risk overfitting, which could lead to catastrophic outcomes when the context changes, such as when performing assistive surgery on a new patient. The ability to generalise and adapt is necessary for robotic skills such as scene understanding and adaptive control and should facilitate robots to operate in volatile (e.g., social) environments (e.g., hospitals) [36,46]. In industrial applications, this allows robots to operate freely while adapting to real world conditions—once the designer has specified preferences over the final outcome.**Safer robots**. AIF agents continuously resolve uncertainty by selecting informative actions that minimise risk [1], which is important for high-stakes, high-uncertainty tasks, such as human-robot interaction [41]. Actions are selected to minimise expected free energy, which minimises risk (expected cost) and ambiguity (expected inaccuracy) [1]. This allows for information seeking behaviour that is accompanied with an explicit and quantifiable measure of risk. Additionally, when uncertain about current states of affair, robots should automatically seek advice and guidance from the user, e.g., via shared control.**Social and collaborative robots**. AIF robots model others’ intentions to predict others’ actions, such as movements [47], enabling intentional understanding [48]. This allows robots to operate safely in social environments by constantly resolving uncertainty about others’ intentions and implicit goals [42]. This embodiment [49] is crucial for social robots, such as personal aides, auxiliary robot nurses and companions, e.g., assisting the disabled and elderly. In collaborative robotics, AIF allows for imitation learning and intentional blending, whence robot goals and intentions can be guided by the user before and during the task [41,50].**Wearable devices**. The belief updating process that underwrites AIF is energetically efficient [51], which should aid the development of wearable devices with a degree of autonomy, such as exoskeletons [52]. This follows as optimising model free energy decreases the movement from prior to posterior, which corresponds to the computational (and hence energetic) cost of inference [1,2]. In addition, wearables directed by human intention [53] should benefit from AIF’s intentional understanding [48], and adaptive and shared control capabilities [41].**Regulatory processes**. Generative models with temporal depth induce allostatic control, whence the robot acts on its environment to pre-empt homeostatic control [54,55]. This should benefit regulatory processes subject to strong external perturbations [16,36], such as closed-loop medical applications such as artificial organs (e.g., the artificial pancreas).**Neurotechnology**. The neurological functional plausibility of specific AIF algorithms [1,46,56] should facilitate integration with the nervous system. This opens new opportunities for neurotechnology, BCI-enabled sensorimotor restoration, perceptual body extension and brain or body enhancement using prosthetics and implants [57]. Currently, AIF provides testable hypotheses for optimising neural excitatory-inhibitory balance using deep brain stimulation to alleviate functional deficits induced by brain lesions [58]. Soon, monitoring of brain activity may predict aberrant neural responses, such as seizures, and anticipate the required intervention.

## 4. Discussion

In this perspective, we explained how active inference—a framework for describing and designing adaptive systems originating in computational neuroscience—can be exploited in robotics. In particular, we surveyed some key features of AIF that could provide solutions to current technical challenges in robotics, and how these could benefit human-centred robotic applications in the short-term.

In brief, the theoretical foundations of AIF suggest the potential for important advances in state-estimation, control, planning and learning that undergirds autonomous robots. This suggests a promising avenue for endowing robots with online adaptive strategies and context-sensitive and explainable decision-making. In turn, these advances could have several applications in robotics, spanning context-adaptive, safe and social robots, wearable devices, regulatory processes and neurotechnology. AIF brings several things to the table in this setting. Perhaps the most important aspects are: (i) a commitment to an explicit, explainable and interpretable world model—in the form of a forward or generative model—that underwrites inference and learning, (ii) framing state estimation, control and planning as different aspects of the same inverse or inference problem, whose solution affords context sensitivity and robustness (iii) and, finally, supplying a tractable objective function that subsumes different kinds of (Bayes) optimality: namely, an expected free energy that subsumes Bayesian decision theory and Bayesian optimal design [2,32]. The latter brings with it a quintessentially belief-based specification of sentient behaviour that can be read as equipping robots with the right kind of curiosity. These foundational features of AIF are, we suppose, also found in human subjects, and therefore place AIF robots in a potentially more empathetic relationship to their human operators. It will be interesting to see whether—or how—these features are leveraged over the next few years.

In short, AIF is generally considered to endow robots and artificial agents with adaptive capabilities. While promising, the application is in its early days and much work remains to be undertaken in order to resolve practical challenges and fulfil the framework’s potential. Current endeavours include scaling AIF to handle high dimensional state-spaces in a variety of applications [10,12,13,59], effectively learning the generative model from data [2,34], and show its practicality in the real world, beyond the lab boundaries. While significant engineering challenges remain, the state-of-the-art laboratory experiments show AIF’s potential as a powerful method in robotics [14].

## Figures and Tables

**Figure 1 entropy-24-00361-f001:**
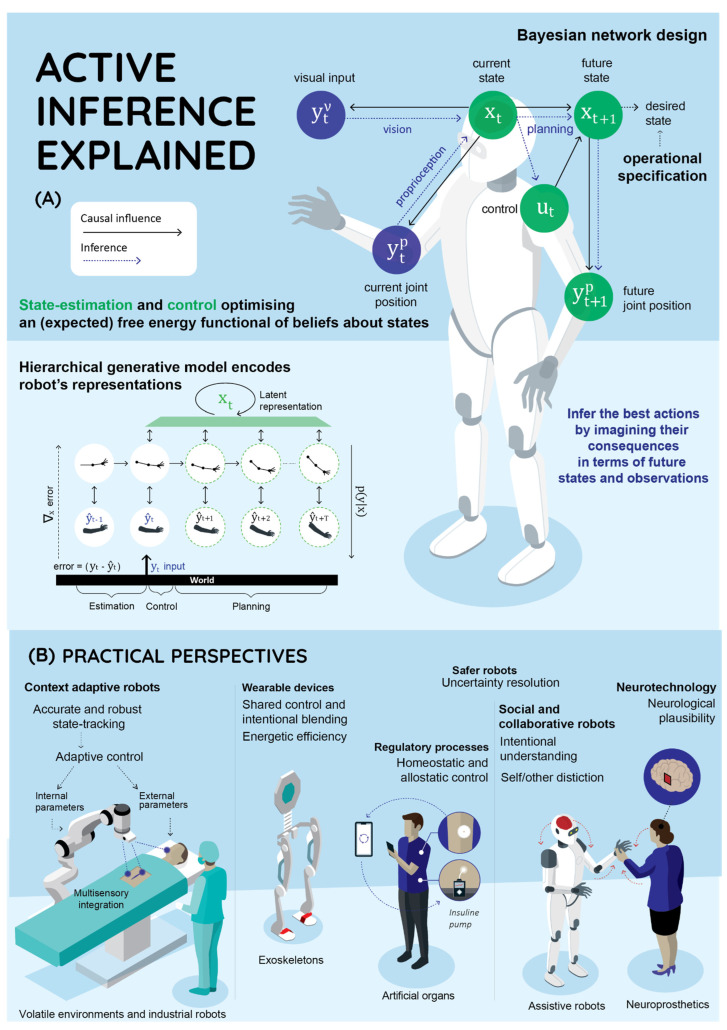
Active inference framework (AIF). AIF could engender important advances in estimation, control, planning and learning in robotics with applications including social, industrial and collaborative robotics, body prosthetics and neurotechnology. (**A**) AIF explained: Blue circles indicate observations while grey circles indicate random variables that need to be inferred. The black arrows indicate causal relationships implicit in a graphical model (e.g., a Bayesian network). The blue arrows indicate the process by which the agent infers future actions and observations. First, the agent infers the current states from available observation modalities (Bayesian fusion). Then, the agent infers the best available course of action by imagining the counterfactual consequences, in terms of future states and observations. These inferential processes are solved by optimising an (expected) free energy functional of beliefs about states and plausible action sequences. AIF generative models may be hierarchical and encode agent’s representations at increasing levels of abstraction and temporal scales. Perception minimises the discrepancy between predictions and input at all levels. The top layer encodes the estimated (and preferred) states of the world—and the bottom layer encodes sensory input. (**B**) Practical perspectives: AIF can provide context sensitivity, online adaptivity, accurate state tracking, uncertainty resolution and shared control in a neurologically plausible fashion throughout a wide range of applications.

## Data Availability

Not applicable.

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
