# Peer review of "How Active Inference Could Help Revolutionise Robotics"

_entropy, 2022, doi:10.3390/e24030361_

Round 1

Reviewer 1 Report

In this paper, the authors give a perspective on potential applications of active inference. The paper is well written and raises some interesting points. Therefore, I recommend it for publication in Entropy.

I only have a few optional remarks that the authors might consider prior to publication:
- Although I believe that the paper, for the most part, is accessible to a relatively broad audience, the introduction is quite technical. I believe that the impact of the paper might increase significantly if the authors were to include a paragraph with a less technical introduction to the basics of active inference.
- The way in which Landauer's principle is mentioned on page 4 seems wrong or misleading to me. Landauer's principle provides a minimal thermodynamic cost to computations and although I can see that, in theory, one can deduce from this that minimal complexity leads to minimal thermodynamic cost,  I do not believe that this is relevant for practical applications, as state-of-art experiments still produce several orders of magnitude more heat than that produced in Landauer's bound.

Reviewer 2 Report

  • I don't quite follow the novelty of this research design and proposal. So I think that this needs to be highlighted clearly early on. Referenced publications propose, in various ways and from different fields, the very same concept of applying Active Inference in robotics, so the unification is I believe, implicit. Hasn't this been discussed before?
  • There are references to equations, such as the forward Kolmogorov equation which are not presented or discussed adequately. Figure 1, visually is sufficient but it should accompany a mathematical formulation and examples.
  • Discussion is rather short and lacks depth. This is also evident in sections 2 and 3 which are rather short as well. I'm not an expert in this field, and I think that I was not given sufficient depth in this area.

Round 2

Reviewer 2 Report

Most of my suggestions were met by the authors.